# Improving follow-up survey completion rates through pilot interventions in the All of Us Research Program: Results from a non-randomized intervention study

**Robert M. Cronin**[1]*, **Xiaoke Feng**[2], **Ashley Able**[3], **Scott Sutherland**[4], **Ben Givens**[4], **Rebecca Johnston**[3], **Charlene Depry**[5], **Katrina W. Le Blanc**[5], **Orlane Caro**[3], **Brandy Mapes**[3], **Josh Denny**[5], **Mick P. Couper**[6,7,8], **Qingxia Chen**[2,9,10], **Irene Prabhu Das**[5]

1 Department of Internal Medicine, The Ohio State University, Columbus, OH, United States of America, 2 Department of Biostatistics, Vanderbilt University Medical Center, Nashville, TN, United States of America, 3 Vanderbilt Institute for Clinical and Translational Research, Vanderbilt University Medical Center, Nashville, TN, United States of America, 4 Vibrent Health, Fairfax, VA, United States of America, 5 All of Us Research Program, Office of the Director, National Institutes of Health, Bethesda, MD, United States of America, 6 Survey Research Center, University of Michigan, Ann Arbor, MI, United States of America, 7 Department of Biostatistics, School of Public Health, University of Michigan, Ann Arbor, MI, United States of America, 8 Institute for Social Research, Survey Research Center, University of Michigan, Ann Arbor, MI United States of America, 9 Department of Biomedical Informatics, Vanderbilt University Medical Center, Nashville, TN, United States of America, 10 Vanderbilt Eye Institute, Vanderbilt University Medical Center, Nashville, TN, United States of America

* robert.cronin@osumc.edu

**Data Availability Statement:** Data used for this manuscript is obtained from the All of Us Research

## Abstract

### Objective

Retention to complete follow-up surveys in extensive longitudinal epidemiological cohort studies is vital yet challenging. *All of Us* developed pilot interventions to improve response rates for follow-up surveys.

### Study design and setting

The pilot interventions occurred from April 27, 2020, to August 3, 2020. The three arms were: (1) telephone appointment [staff members calling participants offering appointments to complete surveys over phone] (2) postal [mail reminder to complete surveys through U.S. Postal Service], and (3) combination of telephone appointment and postal. Controls received digital-only reminders [program-level digital recontact via email or through the participant portal]. Study sites chose their study arm and participants were not randomized.

### Results

A total of 50 sites piloted interventions with 17,593 participants, while 47,832 participants comprised controls during the same period. Of all participants, 6,828 (10.4%) completed any follow-up surveys (1448: telephone; 522: postal; 486: combination; 4372: controls). Follow-up survey completions were 24% higher in the telephone appointment arm than in controls in bivariate analyses. When controlling for confounders, telephone appointment and

Program and not owned by the authors. Data from the All of Us Research Program is accessible only through the Researcher Workbench (https://workbench.researchallofus.org) as stipulated in the informed consent of participants in the program. This data use agreement prohibits investigators from providing row level data on AllofUs participants and thus providing a de-identified dataset is not possible for this manuscript.

**Funding:** This work was supported by awards through the National Institutes of Health (https://allofus.nih.gov/), Office of the Director: Regional Medical Centers: 1 OT2 OD026549; 1 OT2 OD026554; 1 OT2 OD026557; 1 OT2 OD026556; 1 OT2 OD026550; 1 OT2 OD 026552; 1 OT2 OD026553; 1 OT2 OD026548; 1 OT2 OD026551; 1 OT2 OD026555; IAA #: AOD 16037; Federally Qualified Health Centers: HHSN 263201600085U; Data and Research Center: 5 U2C OD023196; The Participant Center: U24 OD023176; Participant Technology Systems Center: 1 U24 OD023163; Communications and Engagement: 3 OT2 OD023205; 3 OT2 OD023206; Community Partners: 1 OT2 OD025277; 3 OT2 OD025315; 1 OT2 OD025337; 1 OT2 OD025276. Pilot Core: 1 OT2 OD023132, 1 OT2 OD023132–02S1, National Heart, Blood, and Lung Institute (https://www.nhlbi.nih.gov/) K23HL141447 (RMC). The funders had no role in study design, data collection and analysis, decision to publish, or preparation of the manuscript.

**Competing interests:** NO authors have competing interests.

combination arms increased rates of completion similarly compared to controls, while the postal arm had no significant effect (odds ratio [95% Confidence Interval], telephone appointment:2.01[1.81–2.23]; combination:1.91[1.66–2.20]; postal:0.92[0.79–1.07]). Although the effects of the telephone appointment and combination arms were similar, differential effects were observed across sub-populations.

## Conclusion

Telephone appointments appeared to be the most successful intervention in our study. Lessons learned about retention interventions, and improvement in follow-up survey completion rates provide generalizable knowledge for similar cohort studies and demonstrate the potential value of precision reminders and engagement with sub-populations of a cohort.

## Introduction

Retention in extensive longitudinal epidemiological cohort studies is vital yet challenging [1]. If attrition is high in these studies, research findings may not be valid. Health surveys are an essential component of longitudinal data collection, yet having participants return to complete follow-up surveys can be difficult [2]. There are multiple reasons why individuals don't fill out follow-up surveys including issues with accessing or submitting the survey, technical issues, not receiving messages, lack of interest, survey being boring or too long, or no time or bad timing [3]. Large cohort studies that used surveys for data collection have had varying response rates, with only a few having online follow-up surveys [4–6]. Prior studies have shown that interventions, such as phone calls, postal mailings, or both have improved follow-up survey completion rates [7–12]. This extensive literature on survey methods shows that multimodal survey methods are more effective than single-mode methods. Survey response rates for electronic, Web-based surveys improve when followed up by letter mailings, phone calls, or both [3], illustrating the importance of multimodal approaches.

The *All of Us* Research Program, or *All of Us*, aims to recruit at least 1 million participants from populations of which 75% will be historically underrepresented in biomedical research and retain them over the many years of this longitudinal program [13,14]. For underrepresented populations, *All of Us* relied on designated definitions of diversity guided by the leading authorities on health disparities and through consultation with a variety of resources and stakeholders as described in our prior work [14]. During a participant's journey, *All of Us* collects multiple sources of information, one of which is health surveys completed by participants through a participant portal [15]. *All of Us* administers health surveys at baseline (i.e., enrollment) and at subsequent time points to collect information on various and timely topics, such as COVID-19. The surveys that participants answered are located here: https://www.researchallofus.org/data-tools/survey-explorer/. The baseline surveys and average completion times in minutes (mean, standard deviation) were: The Basics (6.69,3.48), Lifestyle (2.94, 1.65), and Overall Health (3.1, 1.32). The follow-up surveys and completion times were Health Care Access & Utilization (6.41, 2.72), Personal Health History (7.15, 3.88), and Family Health History (6.61, 3.90). These longitudinal and diverse participant data will inform precision medicine research.

*All of Us* initially relied on only email communication for follow-up activities. Due to the challenges of a digital-only communication approach and retention of diverse populations

[4,6,16], *All of Us* aimed to understand and address the low response rates to follow-up health surveys. While retention can include multiple activities within *All of Us* (e.g., additional consents for genetic return of results and completion of follow-up surveys), our focus in this manuscript is on the retention strategies yielding completion of follow-up surveys. There is a scarcity of both theoretical and practical guidance on crafting optimal surveys for mixed-mode data gathering, such as utilizing both telephone and postal methods [17]. Nevertheless, survey creators often opt for a mixed-mode strategy because it allows for mitigating the drawbacks of each mode while maintaining affordability. By blending a primary method with a secondary, pricier one, researchers can benefit from reduced costs and errors compared to a single-mode approach. Mixed-mode designs entail a deliberate balance between expenses and inaccuracies, particularly addressing non-sampling errors like frame or coverage error, nonresponse error, and measurement error [17]. *All of Us* developed a pilot study that used mixed contact modes and multiple reminders through three retention strategies: 1) telephone appointments, 2) postal mailing, and 3) a combination of telephone appointments and postal mailing. The hypothesis was that completion rates of follow-up surveys would increase in at least one of the retention arms and that the telephone appointment arm would be more effective than the postal arm and less effective than the combination arm. In this manuscript, we describe the methods and results of the pilot study to inform *All of Us'* efforts to increase participant retention by completing the follow-up surveys.

## Methods

### Design of pilot and participating sites

Fig 1 describes the goal and design of the pilot. The pilot period was from April 27, 2020, to August 3, 2020. A subset of the *All of Us* participating clinic sites (N = 50) chose their study arm based on their resource capabilities. Staff at the participating sites were instructed to adhere to the methods for the study (see Standard Operating Procedures appendix). *All of Us* participants are consented when they join the program. Since participants contributed their data towards the *All of Us* Research Program, where consent was obtained for the program, they were aware that their data were going to be used for research purposes. The *All of Us* Institutional Review Board determined that this study did not meet the criteria for human subjects research.

### Identification of participants in the intervention and control groups

The *All of Us* Research Program aims to enroll a diverse group of at least one million people living in the United States, regardless of health status, age, race, ethnicity, sexual orientation, gender identity, or socioeconomic status. The goal is to gather health data from a wide range of individuals to better understand how genetics, lifestyle, environment, and other factors contribute to disease and overall health [13]. Eligible participants comprised those who had consented and completed the three baseline surveys but not **all** three follow-up surveys. Study sites identified the eligible participants for recruitment. Participants from other *All of Us* sites were not included in our analyses because they had been exposed to a retention activity not associated with any of the methods used in this study during the study period.

### Control and interventions

Study sites chose their study arm and participants were not randomized. Participants in all arms and controls were asked to fill out all surveys. The eligible participants from the study sites who were not assigned to the pilot interventions from the study sites made up the control

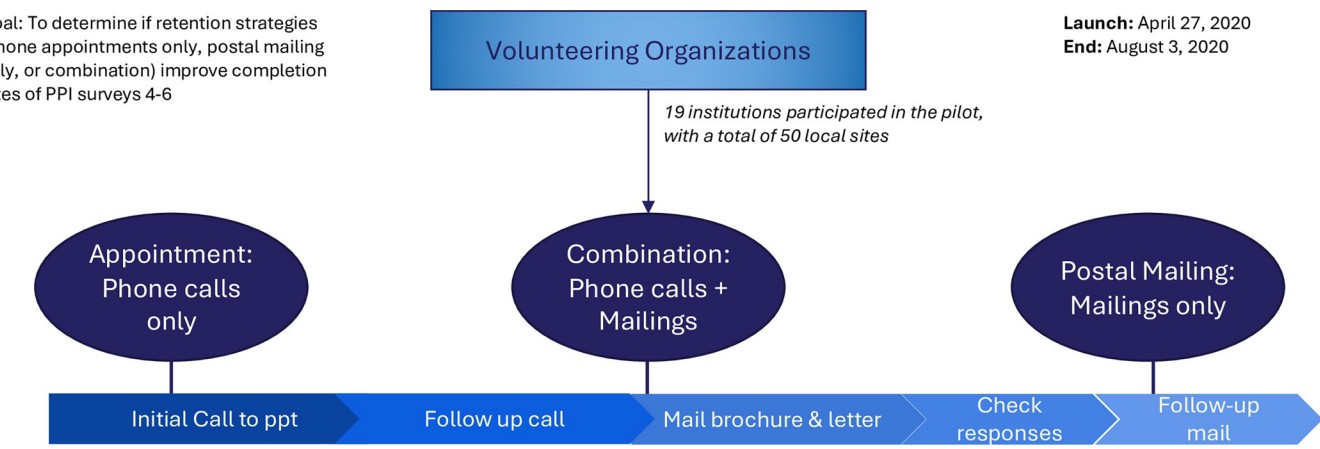

**Fig 1. Goal and design of pilot.** Fifty sites affiliated with All of Us volunteered for one of the three pilot arms. The horizontal bar shows the progression of activities completed at the site level, using the combination arm as an example.

group. The control group also included participants from an additional site who were not exposed to **any** retention pilot or activity. These participants were included to increase the size of the control group and thus allow for more extensive analyses. Program-level digital recontact via email or through the participant portal occurred during the pilot period for all eligible participants, including controls: a recontact campaign for the genetic return of results and another *All of Us* survey, the COVID-19 Participant Experience Survey [18], with additional email reminders.

The **telephone appointment arm** included staff members who made telephone calls to participants who had not completed the surveys, relayed the importance of survey completion, and offered to schedule appointments for these participants to complete the surveys over the phone. The telephone appointment not only reminded individuals about the survey but also offered them the opportunity to fill the surveys out over the phone, like computer assisted telephonic interviews (CATI), which may be more convenient for them. CATI refers to a method of conducting surveys or interviews over the telephone using a computer program to assist interviewers in the process. The software typically helps with tasks such as questionnaire administration, data entry, and management, increasing the efficiency and accuracy of the interview process [19].

The **postal arm** used the U.S. Postal Service to send reminders to participants to complete their surveys. These reminders consisted of a compelling call-to-action letter and survey instruction brochure that reminded participants of the login procedures for the participant portal.

The **combination arm** consisted of an initial contact via a phone call followed by a postal mailing if the participant had not completed a survey within thirty days of the initial call. The initial calls were made at times when staff were able, and not at a specific time period during the pilot implementation. The postal mailing included the Introductory letter and the Survey Instruction Brochure to facilitate portal access for survey completion.

## Data collection

The sites implemented their chosen strategy for at least two months, beginning on or after April 27, 2020. Participating sites received training in the study protocol and data entry procedures. Data for participants were entered into a Research Electronic Data Capture (REDCap) [20] instrument from the sites and obtained from the central Data Research Center's Raw Data Repository, which serves as a central repository for *All of Us* participant survey data. These anonymized collected data were accessed retrospectively for research purposes starting on August 10, 2020.

Study data collected through REDCap included participant ID, number of contacts made to reach participants, appointments scheduled and kept, and if the mail was returned undelivered. Over 100 staff members at study sites entered and monitored their site's study data. Staff used multiple methods for entering data into REDCap, including importing eligible participant identifiers, entering individual participant records manually, and importing all of their data into the central REDCap repository using a local instance of REDCap that included the project data collection instruments.

Data from the Data Research Center's Raw Data Repository included survey completion with the completion date, demographic, and other covariate data for each participant. The demographic data included race and ethnicity, age, sexual orientation and gender, educational attainment, geography (e.g., in a non-urban or urban area), and household income. These demographic data, except for geography, are all self-reported. We compared the results for participants historically represented in biomedical research to those historically underrepresented in biomedical research. The Raw Data Repository data were linked to REDCap data using the participant identification number.

The study data were examined to ensure that all site participant records met the study's eligibility criteria.

## Statistical analysis

The study analysis aimed to determine the following: (1) whether completion rates of at least one follow-up survey increased in one or more study arms; (2) whether at least one follow-up survey completion rates increased more in the telephone appointment arm than the postal arm; and (3) whether at least one follow-up survey completion rates increased more in the combination arm than the other two arms.

The primary outcome of interest is whether participants completed at least one additional follow-up survey during the pilot period. For example, if a participant completed three baseline surveys before the pilot period and then completed one or more follow-up surveys during the pilot period, they were considered to complete additional follow-up surveys. Incremental completion rates were presented for the three study arms and one control arm across three periods (before pilot, during pilot, and two months after pilot), with 95% confidence intervals constructed using the binomial distribution. Note that the completion rates are zeros before pilot by definition. Given the *All of Us* program's emphasis on enrolling underrepresented populations, we also stratified the incremental completion rates by underrepresented (UBR) and represented (RBR) groups [14].

This study is observational with interventions chosen by sites. It is crucial to control for confounding bias in the analysis. We utilized a propensity score (PS) model, which is a logistic regression model for the binary indicator of whether participants received any interventions. Covariates included UBR criteria such as age, sex and gender minority, income, education, geography, race, and ethnicity, as well as the number of missing follow-up surveys. PS is defined as the probability of intervention assignment conditional on observed baseline

characteristics. We compared the PS distribution between intervention arms and the control arm, assessing the overlap of their ranges. As shown in Table 1 and Fig 2, the intervention and control arms are not balanced, and their PS distributions differ, though their PS ranges overlap well.

Various approaches have been proposed in the literature to address confounder bias, including PS matching, PS weighting, PS stratification, and multivariable regression models [21]. The first two estimate the marginal causal effect, while the latter two estimate the conditional treatment effect. For this study, we used multivariable regression models based on the following considerations. (1) Preliminary evidence: Descriptive statistics showed different intervention effects between UBR and RBR groups, highlighting the need to study interactions between intervention arms and characteristics included in the UBR definition variables; (2) Future impact: Moderator effects of participant characteristics are important for future personalized interventions; (3) Available methodology: PS matching, weighting, or stratification were not designed to study interactions between confounders and treatment.

Therefore, we used mixed effects logistic regression model to examine intervention outcomes relative to different covariates and explore interactions to identify the intervention outcomes and effects of select demographic variables. The multivariable regression models included a random intercept for enrollment site to control for site's cluster effect and adjusted for covariates that could have affected the completion of follow-up surveys. These models compared results between each of the intervention and control groups. These covariates included the demographic variables listed above, intervention study arm, Social Security number question missingness and not providing an email address (as proxies for participant engagement), health literacy score from the Brief Health Literacy Scale (which is part of the *All of Us* Overall Health Survey), time of enrollment since *All of Us* initiation (in weeks), number of missing follow-up surveys, and enrollment site size (total number of participants at the site).

We performed additional interaction analyses in the regression models to compare the effects of the different study arms and the following five demographic categories on completion of follow-up surveys with completion rates of the follow-up surveys in the controls: 1) self-reported race/ethnicity, 2) sex and gender, 3) education, 4) geography, and 5) income. We also analyzed completion rates among participants of different age groups, education levels, and racial and ethnic backgrounds.

Statistical analyses were conducted using lme4, forestploter, sjPlot, emmeans, survey, dplyr and ggplot2 packages in R software (version 4.4.0) with significance defined as two-sided $P <$ .05.

## Results

### Descriptive analysis

A total of 50 sites participated. The final study sample included 17,593 participants in the intervention group and 47,832 in the control group (Table 2). Among those in the intervention group, 6,253 participants in the telephone appointment arm, 5,940 in the postal arm and 5,400 in the combination arm. The records of 692 participants who did not meet the study's eligibility criteria or who had withdrawn from *All of Us* were excluded from the analysis.

Of the total number of participants (65,425) in the intervention and control groups, 20,427 (30.9%) were White, 1,628 (2.4%) Asian, 25,615 (39.2%) Black or African American, 11,114 (17%) Hispanic, Latino or Spanish, and 6,641 (10.1%) did not provide racial and ethnic information. There were 10,056 (15.4%) participants had less than high school education, 16,947 (25.9%) with high school education, 16,307 (24.9%) had some college education, 19,384 (29.6%) with a college degree, 2,731 (4.1%) did not provide education information. 23,139

**Table 1. Differences in critical covariates based on pilot arms and controls.**

Summary by study arm.

| | N | Telephone Appointment Pilot N = 6253 | Postal Pilot N = 5940 | Combination N = 5400 | Control N = 47832 | Test Statistic |
|---|---|---|---|---|---|---|
| Sexual and Gender Minority | 65425 | | | | | P<0.0011 |
| Yes | | 0.10 (611) | 0.09 (521) | 0.10 (546) | 0.10 (4996) | |
| No | | 0.90 (5642) | 0.91 (5419) | 0.90 (4854) | 0.90 (42836) | |
| Income | 65425 | | | | | P<0.0011 |
| Less than 25k | | 0.24 (1493) | 0.27 (1622) | 0.53 (2859) | 0.42 (20017) | |
| Greater than or equal to 25k | | 0.76 (4760) | 0.73 (4318) | 0.47 (2541) | 0.58 (27815) | |
| Geography | 65425 | | | | | P<0.0011 |
| Rural | | 0.01 (73) | 0.05 (321) | 0.04 (223) | 0.03 (1651) | |
| Urban | | 0.99 (6180) | 0.95 (5619) | 0.96 (5177) | 0.97 (46181) | |
| Education | 65425 | | | | | P<0.0011 |
| Less than High school | | 0.08 (495) | 0.12 (691) | 0.21 (1153) | 0.16 (7717) | |
| High School | | 0.15 (946) | 0.28 (1640) | 0.33 (1796) | 0.26 (12565) | |
| Some College | | 0.26 (1647) | 0.26 (1518) | 0.22 (1214) | 0.25 (11928) | |
| College degree or more | | 0.49 (3067) | 0.32 (1885) | 0.18 (999) | 0.28 (13433) | |
| No answer | | 0.02 (98) | 0.03 (206) | 0.04 (238) | 0.05 (2189) | |
| Race/Ethnicity | 65425 | | | | | P<0.0011 |
| White | | 0.45 (2843) | 0.34 (1999) | 0.22 (1190) | 0.30 (14395) | |
| Asian | | 0.08 (482) | 0.02 (113) | 0.01 (68) | 0.02 (965) | |
| Black/African American | | 0.14 (887) | 0.31 (1825) | 0.62 (3370) | 0.41 (19533) | |
| Hispanic/Latino/Spanish | | 0.19 (1203) | 0.23 (1386) | 0.07 (359) | 0.17 (8166) | |
| Not provided | | 0.01 (87) | 0.03 (162) | 0.02 (84) | 0.02 (1082) | |
| Other/2 or more races | | 0.12 (751) | 0.08 (455) | 0.06 (329) | 0.08 (3691) | |
| SSN Provided | 65425 | | | | | P<0.0011 |
| Yes | | 0.38 (2405) | 0.31 (1826) | 0.39 (2106) | 0.39 (18819) | |
| No | | 0.62 (3848) | 0.69 (4114) | 0.61 (3294) | 0.61 (29013) | |
| Email Provided | 65425 | | | | | P<0.0011 |
| Yes | | 0.80 (5033) | 0.73 (4341) | 0.69 (3723) | 0.79 (37840) | |
| No | | 0.20 (1220) | 0.27 (1599) | 0.31 (1677) | 0.21 (9992) | |
| Age | 65425 | | | | | P<0.0011 |
| [18, 45) | | 0.36 (2275) | 0.38 (2274) | 0.35 (1893) | 0.35 (16697) | |
| [45, 65) | | 0.36 (2230) | 0.42 (2490) | 0.51 (2743) | 0.46 (22003) | |
| [65, 75) | | 0.18 (1129) | 0.14 (812) | 0.11 (606) | 0.14 (6725) | |
| 75+ | | 0.10 (619) | 0.06 (364) | 0.03 (158) | 0.05 (2407) | |
| Health Literacy Score | 65133 | | | | | P<0.0011 |
| high | | 0.90 (5591) | 0.82 (4841) | 0.75 (4064) | 0.80 (38115) | |
| low | | 0.10 (654) | 0.18 (1051) | 0.25 (1321) | 0.20 (9496) | |
| Time since enrollment | 65425 | 474 602 743 (623 ±197) | 467 558 701 (586 ±158) | 490 583 707 (613 ±183) | 457 615 813 (640 ±222) | P<0.0012 |
| Organization Size | 65425 | 1833 2357 3318 (2407 ±1084) | 1481 9305 9305 (5400 ±3920) | 1960 1960 3736 (3785 ±3617) | 3128 7348 9305 (6552 ±3820) | P<0.0012 |

a b c represent the lower quartile a, the median b, and the upper quartile c for continuous variables.

x ± s represents X ± 1 SD.

N is the number of non-missing values.

Numbers after proportions are frequencies.

Tests used: 1Pearson test; 2Kruskal-Wallis test.

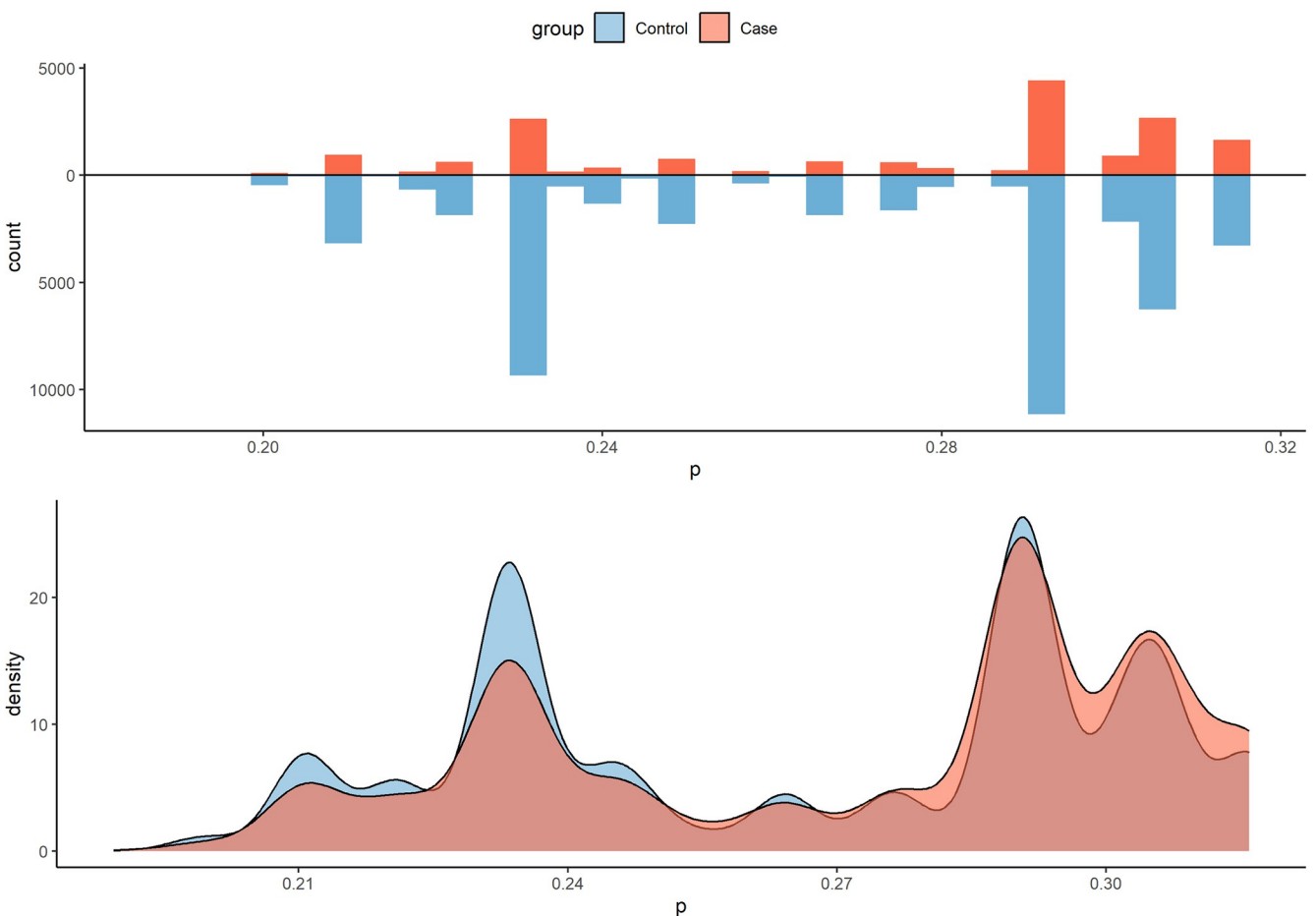

**Fig 2. Propensity score distribution of the pilot arm and controls.**

(35.4%) participants were between age 18–45, 29,466 (45%) between age 45–65, 9272 (14.2%) between age 65–75 and 3,548 (5.4%) participants older than 75. 6,828 (10.4%) completed any of the follow-up surveys, and of these, 6,725 (98.5%) completed all three follow-up surveys.

The follow-up survey completion rate was 24% higher in the telephone appointment arm than in the control group. In addition, there was a ~10% increase in the completion of follow-up surveys among participants in both intervention and control groups (Fig 3). More participants from historically represented groups completed follow-up surveys, regardless of study arm or being a control than participants from historically underrepresented groups (Fig 4).

**Table 2. Number of participants in each study arm in historically underrepresented and represented populations.**

| Study arm | Number of historically underrepresented participants | Number of historically represented participants | Total number of participants in each study arm |
|---|---|---|---|
| Control | 40,967 (85.6%) | 6,865 (14.4%) | 47,832 |
| Appointment | 5,028 (80.4%) | 1,225 (19.6%) | 6,253 |
| Postal | 4,835 (81.4%) | 1,105 (18.6%) | 5,940 |
| Combination | 4,915 (91%) | 485 (9.0%) | 5,400 |
| Total | 55,745 (85.2% of total) | 9,680 (14.7% of total) | 65,425 |

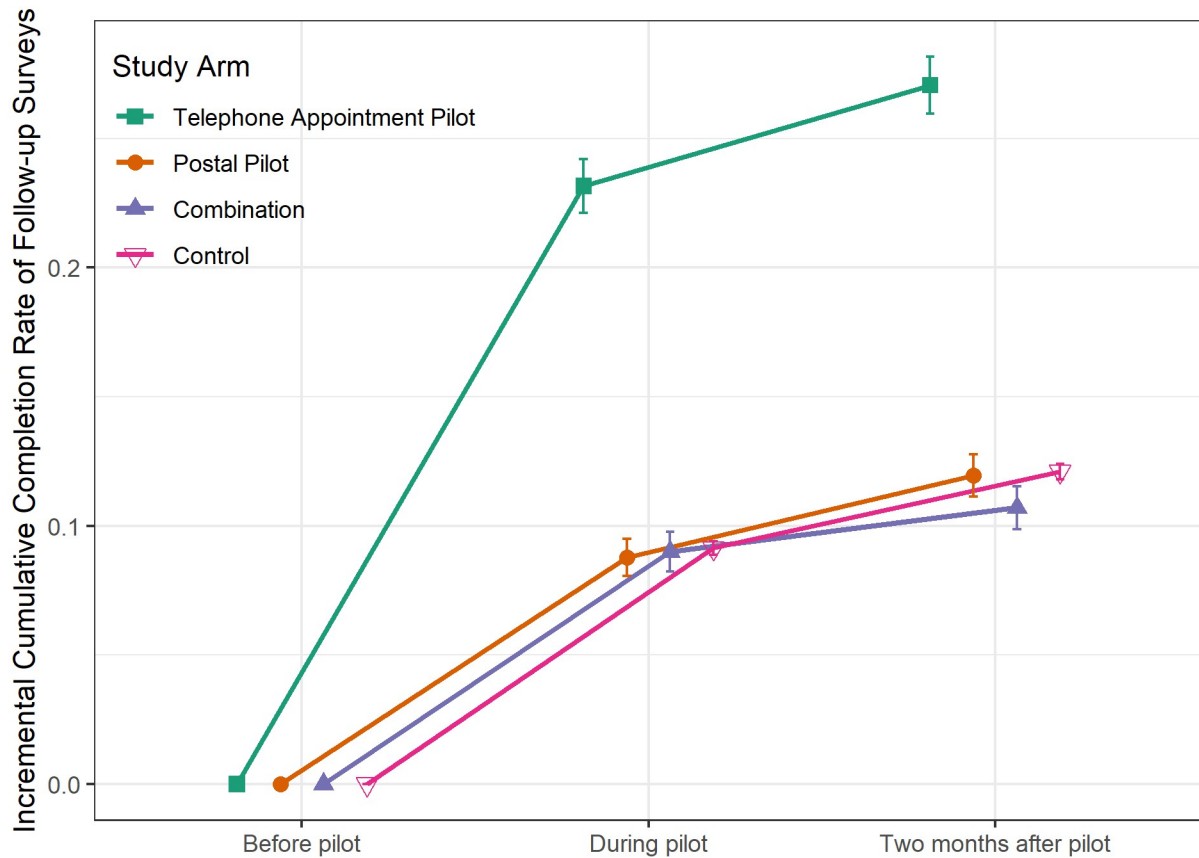

Incremental Cumulative Completion Rate of Follow-up Surveys

| Study arm | Before pilot | During pilot | Two months after pilot |
|---|---|---|---|
| Telephone Appointment Pilot | 0.00 [0.00-0.00] | 0.23 [0.22-0.24] | 0.27 [0.26-0.28] |
| Postal Pilot | 0.00 [0.00-0.00] | 0.09 [0.08-0.10] | 0.12 [0.11-0.13] |
| Combination | 0.00 [0.00-0.00] | 0.09 [0.08-0.10] | 0.11 [0.10-0.12] |
| Control | 0.00 [0.00-0.00] | 0.09 [0.09-0.09] | 0.12 [0.12-0.12] |

**Fig 3. Unadjusted completion rates (proportion) for follow-up surveys by study arm before, during, and after the study period.**

## Multivariable analysis

When we controlled for covariates (Fig 5), the telephone appointment and combination arms were both more likely to complete follow-up surveys than the control arm during the study period in the regression (odds ratio [OR], 2.01; 95% Confidence Interval (CI), 1.81–2.23 for the telephone appointment arm and OR, 1.91; 95% CI, 1.66–2.20 for the combination arm). The postal arm was not significantly different from the control arm (OR, 0.92; 95% CI, 0.79–1.07).

When we considered specific ages, the predicted probability of follow-up survey completion was higher for participants between the ages of 60 and 70 (Fig 6).

For the interaction analysis, the effect of the interventions on the follow-up survey completion rate did not differ significantly by sexual and gender minority status or income. However, the different interventions had different effects on members of different racial and ethnic, education and geography groups (Fig 7A–7C). The telephone appointment arm had higher

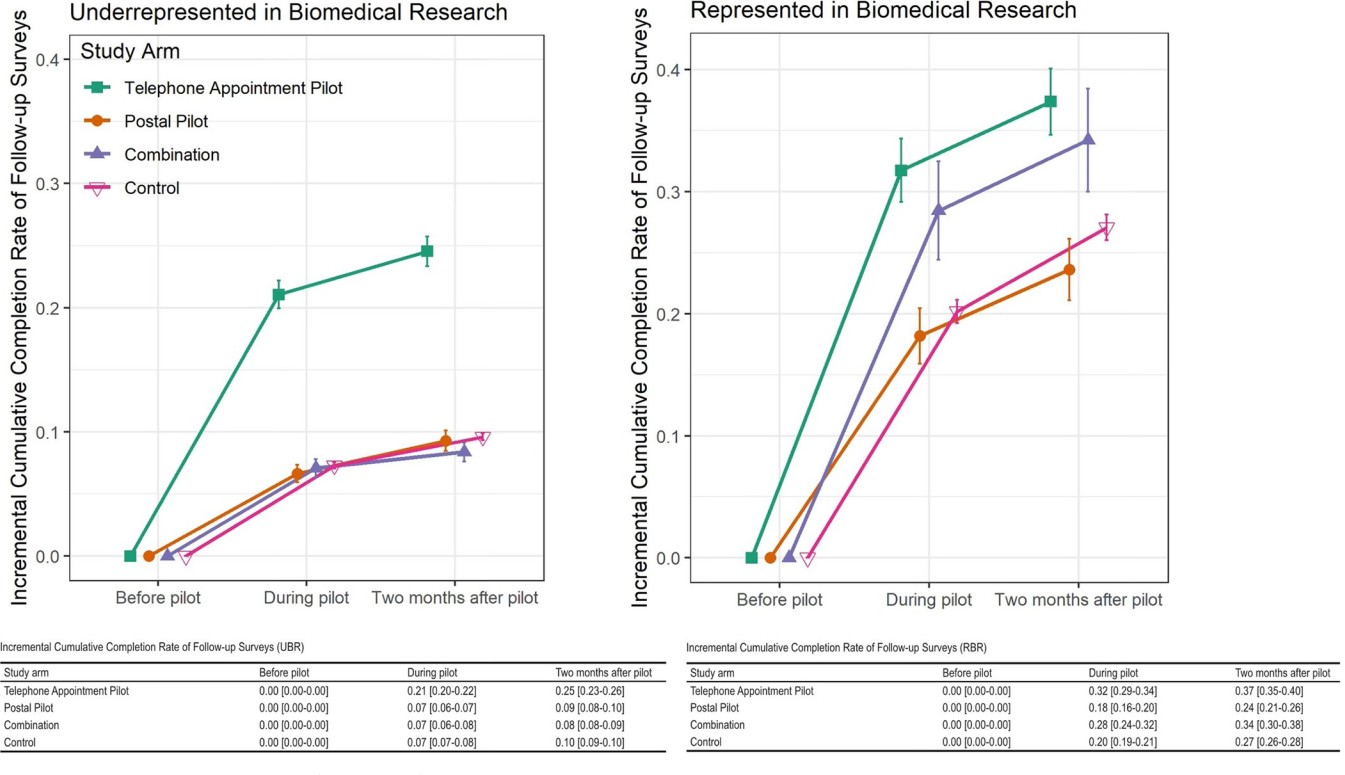

**Fig 4. Unadjusted completion rates (proportion) of follow-up surveys by the study arm in historically underrepresented and represented biomedical research populations.**

follow-up survey completion rates than other arms for all racial and ethnic groups, the postal arm had lower completion rates among Black and African American participants (Fig 7A). Increases in follow-up survey completion rates were more significant among participants who had less than a high school education, high school education and some college education in the combination arm, but survey completion rates increased more among those with a college degree in the telephone appointment arm (Fig 7B). The telephone appointment arm was more effective among participants living in areas with rural and non-rural ZIP codes (Fig 7C). The postal arm was not as effective as the other two arms among participants from rural areas.

## Discussion

Our study reports the effectiveness of three different retention strategies with over 65,000 participants, primarily from diverse populations underrepresented in biomedical research within *All of Us*. While the three interventions and control arm increased the rate of follow-up surveys, the telephone appointment intervention demonstrated the largest increase in bivariate analyses with 24% higher completion than before the pilot study began. When we controlled for covariates, both the telephone appointment and combination arms were similarly effective at increasing completion rates compared to the control arm. The difference between bivariate and multivariable results is likely due to the confounders for the pilot arms and control (Table 1). However, the difference between completion rate increases for the follow-up surveys between the postal and control arms was not statistically significant.

A systematic review by Booker et al. [22] of retention methods in population-based cohort studies described strategies, such as reminders, repeat contacts (phone or mail), and repeated

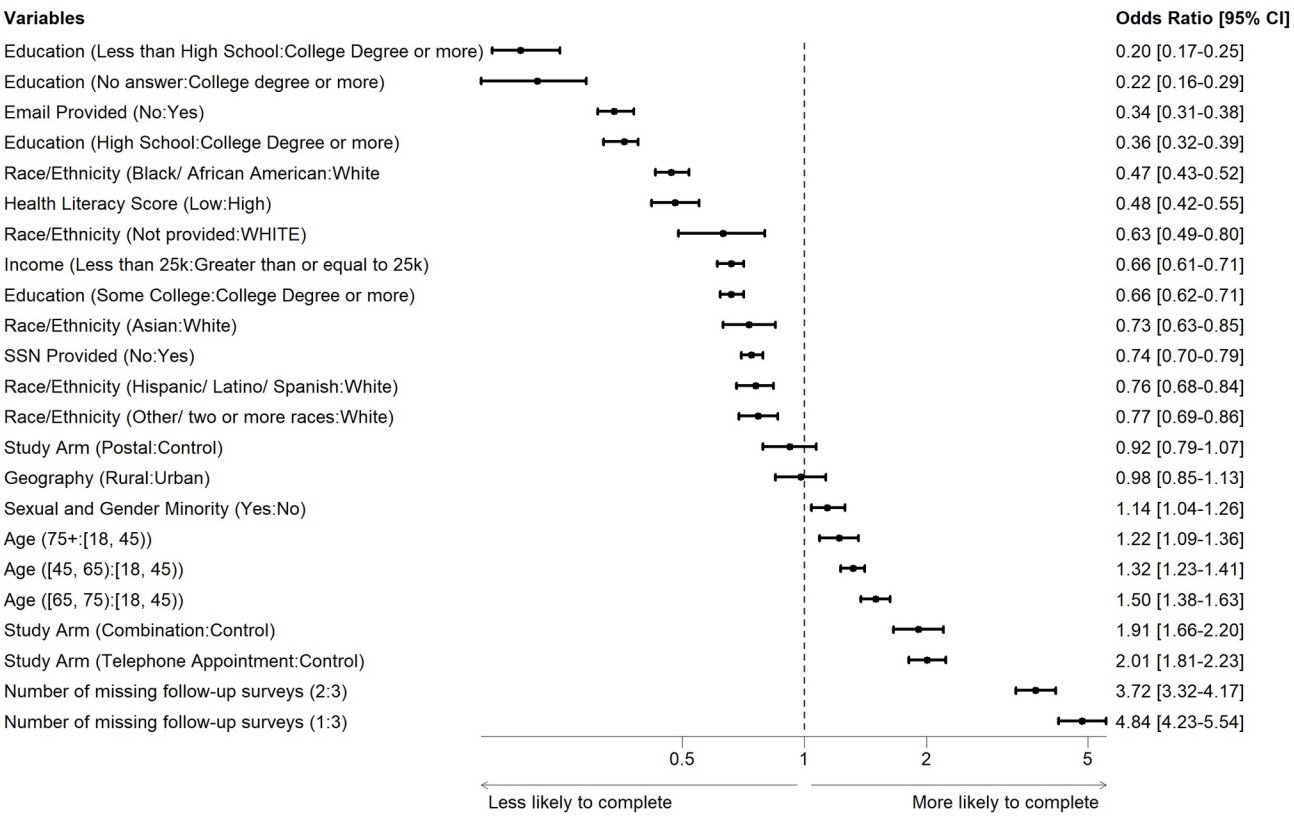

**Fig 5. Multivariable analysis with the covariates of follow-up survey completion.**

mailings of questionnaires were reviewed for 17 studies showing increases in survey response rates of 2%–37%. Higher numbers of reminder letters or postcards led to higher response rates; one study showed increased response rates for mailing a second questionnaire compared with a postcard. Using multiple retention methods including phone calls increased survey response rates by more than 70%. Our results showed increases of up to 24% higher with phone calls. Our lower rates of improvement could be because of the scale of the program or the population that may be more challenging to retain.

Some of the interventions appeared to be more effective in different populations. The telephone appointment intervention was more effective among rural and those with at least an undergraduate college degree than other arms. In contrast, the combination intervention was more effective among those with less than college education. There may be multiple reasons for these differences, in addition to differences in populations. Sites could have spent additional time completing phone calls, whereas, in the combination arm, sites also had to implement a postal component and might not have had as much time to complete additional phone follow-ups. Site bias was also possible where sites could not afford to add postal mailing, chose another arm, or had better connections with their lower education participants leading to higher completion. Differential budget support for retention activities may have contributed to these findings. Some participant characteristics had significant effects on other potential confounders in both intervention and control arms, such as skipping the Social Security number question, not providing an email address, having low health literacy, having a certain age or a particular race or ethnicity, and having a certain level of education. These study findings suggest that *All of Us* should develop and adopt a precision retention framework to

Predicted probabilities of complete additional follow-up surveys

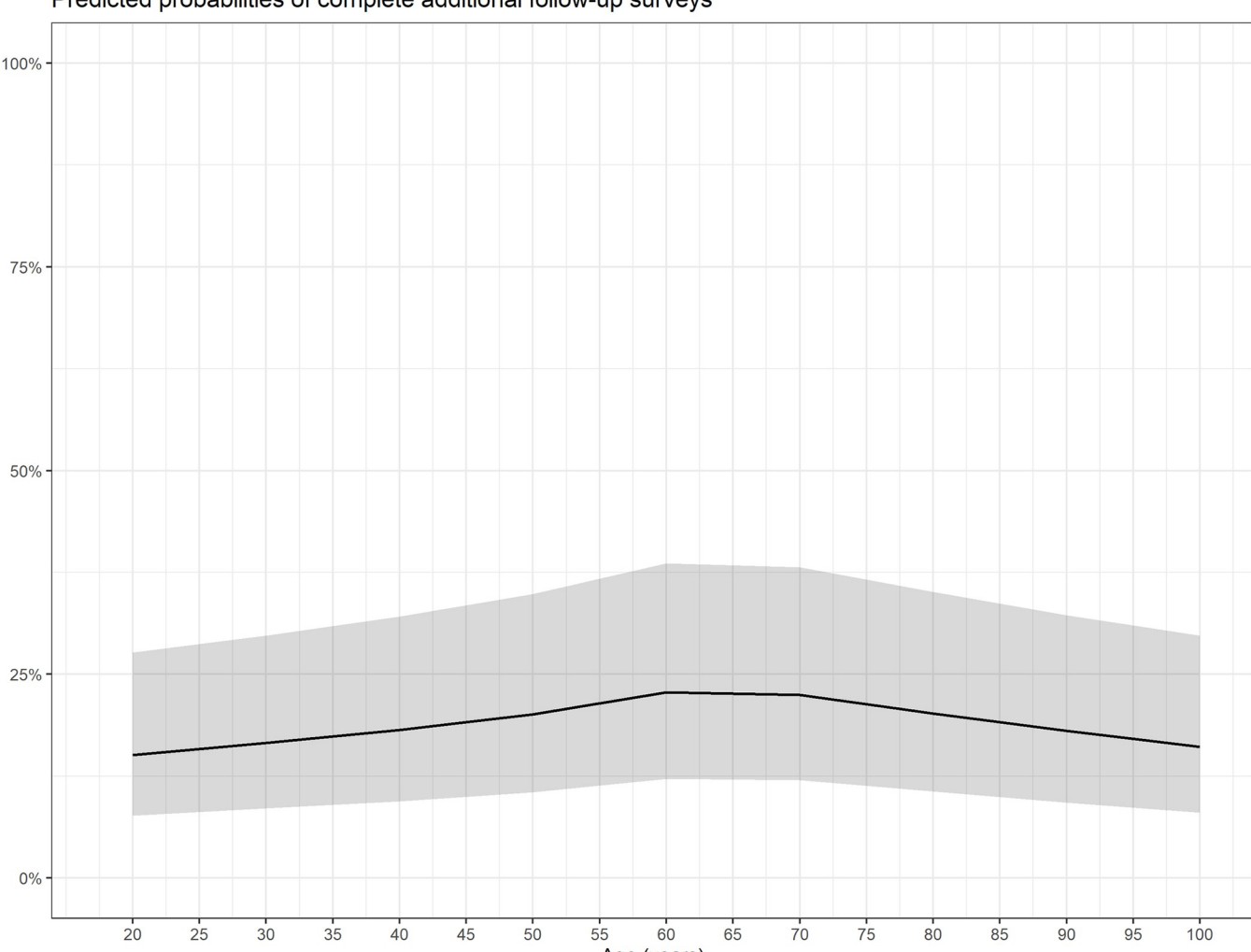

Adjusted for Control study arm, College Degree+, White race, provided SSN, provied Email, all binary UBR variables equal No, three missing follow up surveys, organization size equals 5823, time since enrollment at 631 days

**Fig 6. Odds Ratio with 95% confidence interval (dotted lines) of completion of follow-up surveys by age as a continuous variable.**

strategically direct retention strategies toward participants who would most benefit from that retention strategy, such as phone appointment interventions for African American participants.

Although some of the interventions were particularly effective in certain groups, for other groups, the interventions were no more effective than the control arm. For example, the effects of the different interventions did not differ significantly from that of the control arm among participants with sexual and gender minority status or a lower household income. This could be because of the digital recontact in the control arm. If sites focused on fewer participants who would be less likely to complete additional survey modules because they were less digitally literate, it is possible that more of the digitally literate participants might have completed follow-up surveys in the control arm. The postal intervention was generally less effective than the other two interventions among the historically underrepresented biomedical research populations.

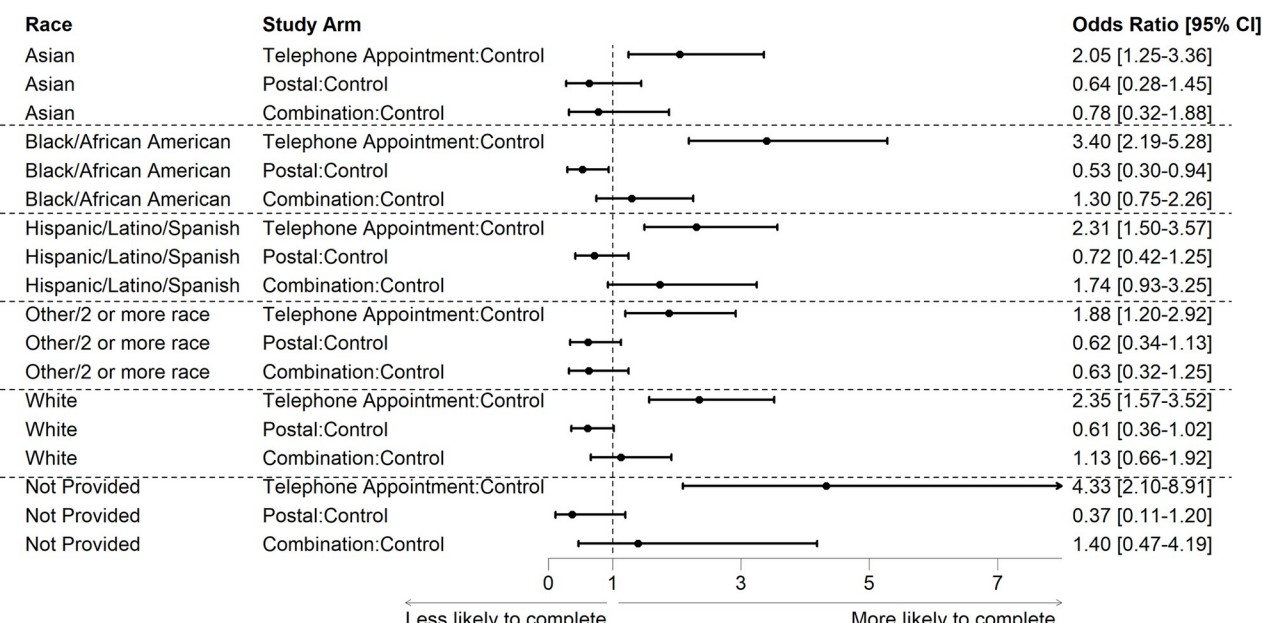

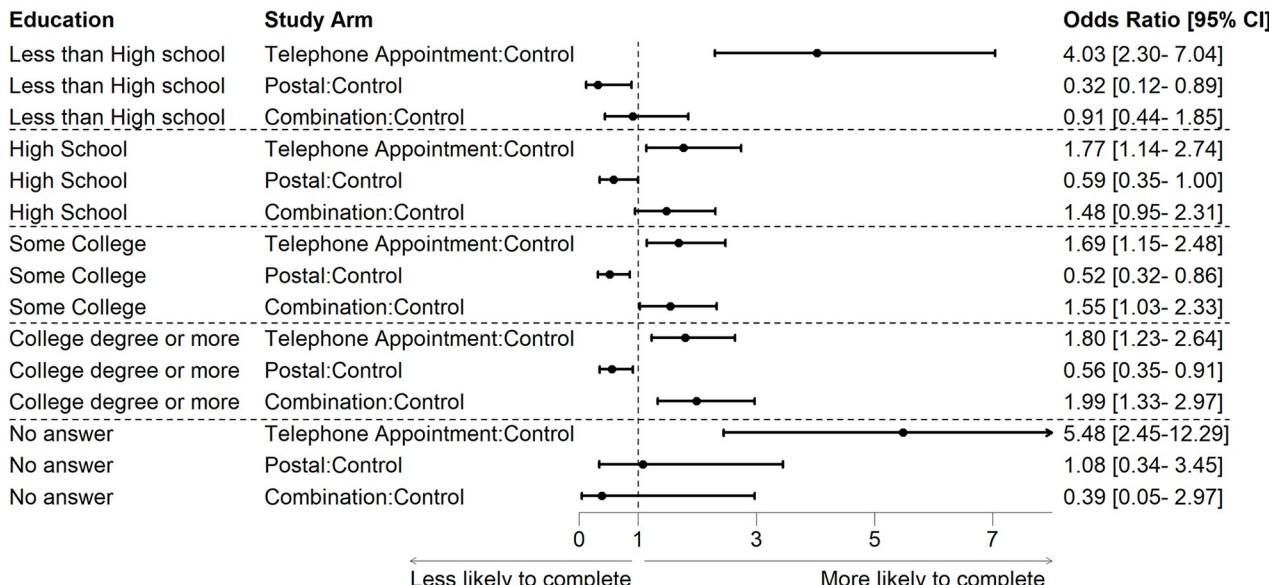

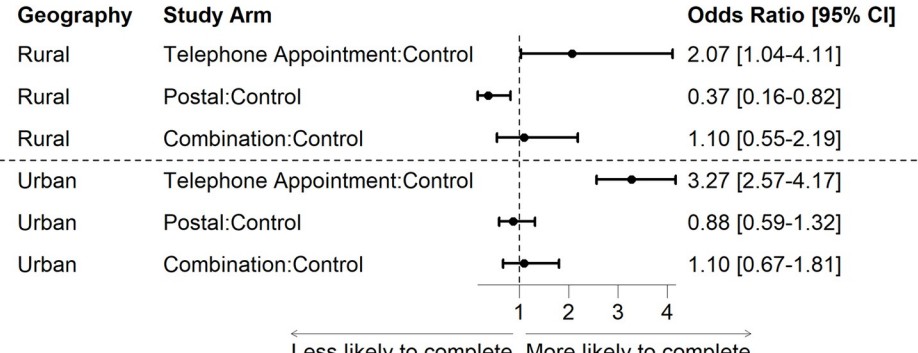

**Fig 7. The interaction analyses for participant characteristics and different pilots.** Differences in completion of surveys in the pilot arms as compared to controls for race/ethnicity are in (a), the highest level of education attained is in (b), and Geography (urban vs. rural) is in (c).

Although the study's interventions helped increase follow-up survey completion rates, *All of Us* still has significant follow-up survey completion challenges. Our pilot interventions align with the literature on survey methods indicating that single-mode survey methods are far less effective than multimode methods [3,7–12] and increased completion rates by up to 25%. In a systematic review by Booker et al. [22] multiple retention methods, including face-to-face interviews, increased the survey completion rate by more than 70%. Future pilot interventions that include face-to-face interviews could improve rates more significantly. Another systematic review by Edwards et al. [23] identified in the 49 trials evaluating the impact of monetary incentives, the odds of responding to traditional mailed surveys doubled when participants received remuneration. *All of Us* partners have begun to pilot incentives to engage participants in follow-up surveys. These incentives varied and included five dollars per follow-up survey completed, thirty dollars for completion of all three follow-up surveys within a four-week period, and different amounts given to complete them within certain timeframes.

Anecdotally, we learned that participants may have experienced challenges when attempting to complete outstanding follow-up surveys using a digital-only strategy for study reminders, including technical challenges that were participant-based or related to the participant portal. Digital-only communication strategies in large, diverse populations traditionally underrepresented in biomedical research are insufficient to motivate survey completion. *All of Us* conducted a year-long pilot using Computer-Assisted Telephone Interviews (CATI) as an additional modality to facilitate survey completion among participants [19]. Findings showed that use of CATI increased survey completion for the *All of Us*. Given the evidence of interest and success, especially among UBR participants, the Program has incorporated CATI as an evidence-based, multimodal strategy for survey completion.

Limitations caution interpretation of this study. First, the study did not randomly assign participants to intervention or control arms (sites chose the intervention arm and which participants to contact as part of the study). We used propensity scores to demonstrate that after adjusting for the covariates we evaluated, the likelihood of assigning to the intervention and control groups overlapped well. Using a randomized controlled design within this real-world context was not feasible because sites had to continue implementing other retention activities during the study period to meet *All of Us* requirements. However, future evaluation of retention strategies will benefit from random assignment. Second, sites could have implemented other retention interventions simultaneously with the study interventions. We described standard operating procedures for these interventions, and sites agreed to hold on to other retention activities during the study period; however, we could not confirm if this was the case. Third, selection bias may be present as some sites opted to participate in the study while others did not. Fourth, we assumed that REDCap entries included records of all attempted participant contacts and that follow-up survey completion was recorded regardless of whether contact attempts were successful. Since this study used an intent-to-treat analysis, the results could have led to a conservative estimate of the actual effect. Finally, other potential confounders, including digital literacy level, access to digital technologies, and specific site characteristics, could not be explored in these analyses. Analyzing these data could lead to further insights into the effect of these strategies.

The lessons learned from this study about retention interventions and improvement in follow-up survey completion rates provide generalizable knowledge for other similar cohort studies and demonstrate the potential value of precision reminders and engagement with sub-

populations of a cohort. *All of Us* will use the findings from this pilot study to improve retention approaches and to offer tested strategies, including systematic tracking and site-level adaptations, to *All of Us* consortium members. In addition, the lessons learned from the real-world implementation of the study interventions (including lessons about costs, staffing, and other resources needed) from this pilot study will contribute to the design of future enrollment and retention pilot interventions for *All of Us* and other similar cohort studies. *All of Us* will also need to assess the generalizability, scalability, and cost of each retention strategy and place those factors into the context of their effect on follow-up survey completion rates. Other sites or other sizeable longitudinal cohort programs that choose to adopt one of the three interventions used in this study could assess if they have similar results to those in the study and which factors contribute to the success or failure of these interventions in local contexts.

## Acknowledgments

We wish to thank our participants who have joined *All of Us* and contributed to the surveys, helped refine early materials, engaged in developing and evaluating the surveys, and provided other ongoing feedback. We thank the countless co-investigators and staff across all awardees and partners, without which *All of Us* would not have achieved our current goals.

We also thank the following NIH staff who provided their expertise in the pilot analysis: Chris Foster, Sarra Hedden, and Tamara Litwin. In addition, we would like to recognize the contributions of the project planning team of *All of Us* investigators, site managers, and NIH staff in designing pilot strategies and the standardized protocol and data collection before implementation.

*All of Us Survey Committee Members*: James McClain, Brian Ahmedani, Michael Manga-niello, Kathy Mazor, Heather Sansbury, Alvaro Alonso, Sarra Hedden, Randy Bloom

Pilot Core at the DRC: Cassie Springer, Ashley Able, Ryan Hale, and Lina Suileman

We also wish to thank *All of Us* Research Program Director Josh Denny, Holly Garriock, Stephanie Devaney, and our partners Vibrent, Scripps, and Leidos.

"Precision Medicine Initiative, PMI, All of Us, the *All of Us* logo, and The Future of Health Begins with You are service marks of the US Department of Health and Human Services."

## Author Contributions

**Data curation:** Xiaoke Feng, Scott Sutherland, Ben Givens.

**Visualization:** Xiaoke Feng.

**Writing – original draft:** Robert M. Cronin, Brandy Mapes, Josh Denny, Mick P. Couper, Qingxia Chen, Irene Prabhu Das.

**Writing – review & editing:** Ashley Able, Scott Sutherland, Ben Givens, Rebecca Johnston, Charlene Depry, Katrina W. Le Blanc, Orlane Caro, Brandy Mapes, Josh Denny, Mick P. Couper, Qingxia Chen, Irene Prabhu Das.

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
