## [Decision Letter · Decision Letter 0]

16 Apr 2024

PONE-D-23-28732Improving Follow-Up Survey Completion Rates through Pilot Interventions in the All of Us Research ProgramPLOS ONE

Dear Dr. Cronin,

Thank you for submitting your manuscript to PLOS ONE. After careful consideration, we feel that it has merit but does not fully meet PLOS ONE’s publication criteria as it currently stands. Therefore, we invite you to submit a revised version of the manuscript that addresses the points raised during the review process.

**ACADEMIC EDITOR: Please follow all suggestions by the reviewers (particularly Reviewer 2) and write a fully itemised response. **

We look forward to receiving your revised manuscript.

Kind regards,

Christian von Wagner

Academic Editor

PLOS ONE

Reviewers' comments:

Reviewer's Responses to Questions

**Comments to the Author**

1. Is the manuscript technically sound, and do the data support the conclusions?

Reviewer #1: Yes

Reviewer #2: No

2. Has the statistical analysis been performed appropriately and rigorously? 

Reviewer #1: Yes

Reviewer #2: Yes

3. Have the authors made all data underlying the findings in their manuscript fully available?

Reviewer #1: Yes

Reviewer #2: Yes

4. Is the manuscript presented in an intelligible fashion and written in standard English?

Reviewer #1: Yes

Reviewer #2: Yes

5. Review Comments to the Author

Reviewer #1: I appreciate the opportunity to review the manuscript titled “Improving Follow-Up Survey Completion Rates through Pilot Interventions in the All of Us Research Program" for PLOS ONE

The manuscript presents a reliable and comprehensive study to improve response rates for follow-up surveys. This study compared the response rates of three intervention group including telephone appointments, postal mail, and a combination of the two with control group that only received digital reminders. The study provides evidence that subpopulations have different response rates and reminders affect each subpopulation differently.

The manuscript is reliable both in terms of participant number and methodology, which makes it worth publishing.

The only suggestion I have is to provide a little bit of information about the base-line and follow up questions and how much time does it take from participants.

Reviewer #2: The study describes a non-randomised study of interventions that compared three inventions aimed at increasing response rates for follow-up surveys in the all of us research program. The interventions consisted of postal or phone reminders. The main problem of the study is that individuals were not randomly allocated to the conditions limiting so the interpretation of the results due to the potential biases. Additionally, the introduction, methods and discussion sections miss relevant literature. There are no previous studies mentioned that tested reminders or the effect of offering an alternative participation modality. The results of the study are also not discussed in line with previous studies.

Title:

• The title should state that is not a randomised study. Something along the lines of: “Improving Follow-Up Survey Completion Rates through Pilot Interventions in the All of Us Research Program: Results from a non-randomised intervention study”

Abstract:

• In general, the abstract is very short and missing some important information. The limit for the word count should be 300 words, but the authors only used 188. The missing information were:

o Telephone appointment intervention could be briefly explained in brackets. You should mention that it consists of offering an alternative participation modality.

o The digital online reminder could also be explained. It sounds like an email reminder.

o If the word count allows it, it would be great to get some information about the allocation to the conditions (stating that it was not random).

o Absolute completion rates should be communicated. What was the baseline?

o The conclusion sounds very general and only highlights that the study improved understanding of how to improve retention rates. Nothing was said about the implications that telephone appointments were the only successful intervention (alone or in combination with the postal reminder

Introduction:

• The introduction section is very short and missed some information from previous literature on why individuals don’t fill out follow-up surveys.

• There is no rational for the tested interventions. There surely must been studies that have tested these interventions beforehand.

• How do you define the underrepresented population? What are the characteristics?

• It would also have been helpful to get some information/theory/model about what the interventions target specifically (e.g. what is the rational that telephone appointments should be more effective that postal reminder)? They both seem to address the problem of procrastination, but the phone is more difficult to avoid.

Method

• The methods section is also very short and should feature more information about the study design.

o The study nature (non-randomised intervention study) should be mentioned in the design section.

o I think that the telephone appointment intervention is kind of complex and requires some more details as it does not only remind individuals about the survey but also offers them to fill it out over the phone (CATI), which may be more convenient for them. Some literature on the perception of CATI would help the reader to better understand that it was not just a reminder, but the offer of a different modality.

o The combination arm should also be explained in more detail. What were the timepoints of the two interventions? Were individual first sent the letter and then called? If yes, then the letter serves a primer. This should be acknowledged.

o Some information about the study population would be welcome. Apart from the sentence in the introduction about the All of Us study, nothing is said about the characteristics of the study population, such as age range, …

o What happened to study participants, who missed more than one follow-up survey. Where they asked to fill out just one or all of them?

Analysis section

o You mentioned that study participants needed to have missed at least one follow-up survey to qualify for the intervention, thus the outcome measure is completing any missing follow-up survey. The completion rate is therefore just reefing to completing at least one follow-up. This needs to be described clearly.

o Could you also include the number of missing follow-up surveys as a covariate in the regression model? It would be nice to see if the interventions’ impact is associated with that.

o Did the regression control for that clustering of the intervention? In the method section it is written, that individuals were allocated to arms according to the preferences of the sites. You should control for this.

Results:

• Please include a table of the characteristics of the study sample according to the conditions and also conduct some univariate tests to see if there are differences in the allocation. This is to check for selection biases. I believe that you put it in the appendix (table 1). If this is the case, then mention the results of the tests, it does not seems as if the conditions were balanced. Please also refer to that table in the results section and not just the discussion.

• Provide also some descriptive statistics about the characteristics of the overall study participants in text form

• Mention the completion rates in absolute terms in the results for all conditions. The figures are of low resolution and difficult to read. You could easily translate them into tables to make it easier to understand.

• What was the rational for the interaction analysis? If this is based on findings from previous literature, then you should add a reference in the introduction section and state that you want to look at it too.

Discussion

• Overall the discussion is missing relevant literature and the implications of the imbalances in the conditions should be discussed in more detail. For me the simple regression does not make much sense in this context and could be removed completely.

o You could add some links to findings from previous studies that tested similar interventions.

o How do your results align with them and how does your study differ in terms of interventions, study population and context.

• Can elaborate on this statement: “If sites focused on fewer participants who would be less likely to complete additional modules because they were less digitally literate, the digitally literate participants might have completed follow-up surveys in the control arm.”

• Can you cite studies that tested postal reminders for online survey and what they found?

• Can you say something more about including incentives? Are they monetary, conditional, lottery-based,…

• You should mention propensity score in the statistical analysis section already.

6. PLOS authors have the option to publish the peer review history of their article (what does this mean?). If published, this will include your full peer review and any attached files.

Reviewer #1: **Yes: **Nima Ghahari

Reviewer #2: No

---

## [Author Response · Author response to Decision Letter 0]

11 Jun 2024

Reviewer Comments:

Reviewer #1: I appreciate the opportunity to review the manuscript titled “Improving Follow-Up Survey Completion Rates through Pilot Interventions in the All of Us Research Program" for PLOS ONE

The manuscript presents a reliable and comprehensive study to improve response rates for follow-up surveys. This study compared the response rates of three intervention group including telephone appointments, postal mail, and a combination of the two with control group that only received digital reminders. The study provides evidence that subpopulations have different response rates and reminders affect each subpopulation differently.

The manuscript is reliable both in terms of participant number and methodology, which makes it worth publishing.

>>>>>>>>

Response: Thank you very much.

>>>>>>>>

The only suggestion I have is to provide a little bit of information about the base-line and follow up questions and how much time does it take from participants.

>>>>>>>>

Response: We added in the website where the questions are located to the Introduction:

The questions that participants answered are located here: https://www.researchallofus.org/data-tools/survey-explorer/. 

We also added the following for mean (SD) completion times in the results

The baseline surveys and average completion times in minutes (mean, standard deviation) were: The Basics (6.69,3.48), Lifestyle (2.94, 1.65), and Overall Health (3.1, 1.32). The follow-up surveys and completion times were Health Care Access & Utilization (6.41, 2.72), Personal Health History (7.15, 3.88 ), and Family Health History (6.61, 3.90).

>>>>>>>>

Reviewer #2: The study describes a non-randomised study of interventions that compared three inventions aimed at increasing response rates for follow-up surveys in the all of us research program. The interventions consisted of postal or phone reminders. The main problem of the study is that individuals were not randomly allocated to the conditions limiting so the interpretation of the results due to the potential biases. Additionally, the introduction, methods and discussion sections miss relevant literature. There are no previous studies mentioned that tested reminders or the effect of offering an alternative participation modality. The results of the study are also not discussed in line with previous studies.

Title:

• The title should state that is not a randomised study. Something along the lines of: “Improving Follow-Up Survey Completion Rates through Pilot Interventions in the All of Us Research Program: Results from a non-randomised intervention study”

>>>>>>>>

Response: We changed the title to the following: 

Improving Follow-Up Survey Completion Rates through Pilot Interventions in the All of Us Research Program: Results from a non-randomized intervention study 

>>>>>>>>

Abstract:

• In general, the abstract is very short and missing some important information. The limit for the word count should be 300 words, but the authors only used 188. The missing information were:

o Telephone appointment intervention could be briefly explained in brackets. You should mention that it consists of offering an alternative participation modality.

o The digital online reminder could also be explained. It sounds like an email reminder.

>>>>>>>> 

Response: We added a brief description of the telephone intervention, postal, and digital only reminder in brackets:

The three arms were: (1) telephone appointment [staff members calling participants offering appointments to complete surveys over phone] (2) postal [mail reminder to complete surveys through U.S. Postal Service], and (3) combination of telephone appointment and postal. Controls received digital-only reminders [program-level digital recontact via email or through the participant portal].

>>>>>>>>

o If the word count allows it, it would be great to get some information about the allocation to the conditions (stating that it was not random).

>>>>>>>>

Response: We added the following about allocation to study arms:

Study sites chose their study arm and participants were not randomized

>>>>>>>>

o Absolute completion rates should be communicated. What was the baseline?

>>>>>>>>

Response: We added the absolute completion rates of follow-up surveys. All of these participants completed baseline surveys.

Of all participants, 6,828 (10.4%) completed any follow-up surveys (1448 : telephone; 522: postal; 486: combination; 4372: controls). Follow-up survey completions were 24% higher in the telephone appointment arm than in controls in bivariate analyses. 

>>>>>>>>

o The conclusion sounds very general and only highlights that the study improved understanding of how to improve retention rates. Nothing was said about the implications that telephone appointments were the only successful intervention (alone or in combination with the postal reminder

>>>>>>>>

Response: We added that telephone appointments were the most successful intervention to the conclusions. 

Telephone appointments appeared to be the most successful intervention in our study. 

>>>>>>>>

Introduction:

• The introduction section is very short and missed some information from previous literature on why individuals don’t fill out follow-up surveys.

>>>>>>>> 

Response: We added some information from previous literature and why individuals don’t fill out follow-up surveys

There are multiple reasons why individuals don’t fill out follow up surveys including issues with accessing or submitting the survey, technical issues, not receiving messages, lack of interest, survey being boring or too long, or no time or bad timing[1].

>>>>>>>> 

• There is no rationale for the tested interventions. There surely must been studies that have tested these interventions beforehand.

>>>>>>>> 

Response: We clarified this in our introduction as these interventions have been effective in prior studies.

Prior studies have shown that interventions, such as phone calls, postal mailings, or both have improved follow-up survey completion rates[2-7]. This extensive literature on survey methods shows that multimodal survey methods are more effective than single-mode methods.

>>>>>>>> 

• How do you define the underrepresented population? What are the characteristics?

>>>>>>>> 

Response: the definition is described in our prior work[8]:

However we added a sentence that further describes the definition and added the reference:

For underrepresented populations, All of Us relied on designated definitions of diversity guided by the leading authorities on health disparities and through consultation with a variety of resources and stakeholders as described in our prior work[8]

>>>>>>>> 

• It would also have been helpful to get some information/theory/model about what the interventions target specifically (e.g. what is the rational that telephone appointments should be more effective that postal reminder)? They both seem to address the problem of procrastination, but the phone is more difficult to avoid.

>>>>>>>> 

Response: We added the following to the manuscript

There is a scarcity of both theoretical and practical guidance on crafting optimal surveys for mixed-mode data gathering, such as utilizing both telephone and postal methods[9]. Nevertheless, survey creators often opt for a mixed-mode strategy because it allows for mitigating the drawbacks of each mode while maintaining affordability. By blending a primary method with a secondary, pricier one, researchers can benefit from reduced costs and errors compared to a single-mode approach. Mixed-mode designs entail a deliberate balance between expenses and inaccuracies, particularly addressing non-sampling errors like frame or coverage error, nonresponse error, and measurement error[9].

>>>>>>>> 

Method

• The methods section is also very short and should feature more information about the study design.

o The study nature (non-randomised intervention study) should be mentioned in the design section.

>>>>>>>> 

Response: We added the following to the design section:

Study sites chose their study arm and participants were not randomized.

>>>>>>>> 

o I think that the telephone appointment intervention is kind of complex and requires some more details as it does not only remind individuals about the survey but also offers them to fill it out over the phone (CATI), which may be more convenient for them. Some literature on the perception of CATI would help the reader to better understand that it was not just a reminder, but the offer of a different modality.

>>>>>>>> 

Response: We clarified the telephone appointment intervention with the following:

The telephone appointment not only reminded individuals about the survey but also offered them the opportunity to fill the surveys out over the phone, like computer assisted telephonic interviews (CATI), which may be more convenient for them. CATI refers to a method of conducting surveys or interviews over the telephone using a computer program to assist interviewers in the process. The software typically helps with tasks such as questionnaire administration, data entry, and management, increasing the efficiency and accuracy of the interview process [10].

>>>>>>>> 

o The combination arm should also be explained in more detail. What were the timepoints of the two interventions? Were individual first sent the letter and then called? If yes, then the letter serves a primer. This should be acknowledged.

>>>>>>>> 

Response: We clarified the combination arm as follows:

The combination arm consisted of an initial contact via a phone call followed by a postal mailing if the participant had not completed a survey within thirty days of the initial call. The initial calls were made at times when staff were able, and not at a specific time period during the pilot implementation. The postal mailing included the Introductory letter and the Survey Instruction Brochure to facilitate portal access for survey completion. 

>>>>>>>> 

o Some information about the study population would be welcome. Apart from the sentence in the introduction about the All of Us study, nothing is said about the characteristics of the study population, such as age range, …

>>>>>>>> 

Response: We added the following verbiage and reference for the all of us study population to the “Identification of participants in the Intervention and Control Groups” section. 

The All of Us Research Program aims to enroll a diverse group as described in the introduction of at least one million people living in the United States, regardless of health status, age, race, ethnicity, sexual orientation, gender identity, or socioeconomic status. The goal is to gather health data from a wide range of individuals to better understand how genetics, lifestyle, environment, and other factors contribute to disease and overall health[11].

>>>>>>>> 

o What happened to study participants, who missed more than one follow-up survey. Were they asked to fill out just one or all of them?

>>>>>>>> 

Response: Participants were asked to fill out all surveys. We added this to the “Controls and Interventions” section:

Participants in all arms and controls were asked to fill out all surveys.

>>>>>>>> 

Analysis section

o You mentioned that study participants needed to have missed at least one follow-up survey to qualify for the intervention, thus the outcome measure is completing any missing follow-up survey. The completion rate is therefore just reefing to completing at least one follow-up. This needs to be described clearly.

>>>>>>>> 

Response: For clarity we clarified and moved the following to the beginning of the paragraph of the analysis section: 

The study analysis aimed to determine the following: (1) whether completion rates of at least one follow-up survey increased in one or more study arms; (2) whether at least one follow-up survey completion rates increased more in the telephone appointment arm than the postal arm; and (3) whether at least one follow-up survey completion rates increased more in the combination arm than the other two arms.

>>>>>>>> 

o Could you also include the number of missing follow-up surveys as a covariate in the regression model? It would be nice to see if the interventions’ impact is associated with that.

>>>>>>>> 

Response: Yes, we made this modification and added this covariate to the models. The methods were updated as below as were results and figures to include these covariates: 

These covariates included the demographic variables listed above, intervention study arm, Social Security number question missingness and not providing an email address (as proxies for participant engagement), health literacy score from the Brief Health Literacy Scale (which is part of the All of Us Overall Health Survey), time of enrollment since All of Us initiation (in weeks), number of missing follow-up surveys, and enrollment site size (total number of participants at the site).

>>>>>>>> 

o Did the regression control for that clustering of the intervention? In the method section it is written, that individuals were allocated to arms according to the preferences of the sites. You should control for this.

>>>>>>>> 

Response: In this revision, the site was incorporated into the model as a random effect and results were updated to reflect the change.

>>>>>>>> 

Results:

• Please include a table of the characteristics of the study sample according to the conditions and also conduct some univariate tests to see if there are differences in the allocation. This is to check for selection biases. I believe that you put it in the appendix (table 1). If this is the case, then mention the results of the tests, it does not seems as if the conditions were balanced. Please also refer to that table in the results section and not just the discussion.

>>>>>>>> 

Response: We agree. The conditions were not balanced in the univariate analysis, and we have incorporated this observation into the Results section. As per your suggestion in the subsequent comment, we have included detailed information regarding the propensity score model in the Method section, along with a figure comparing the propensity scores of the intervention and control arms, now situated in the Results section. Our findings indicate that although there are discrepancies in the balance of individual conditions between the intervention and control arms, their propensity score distributions exhibit significant overlap. Consequently, we assert that a multivariable regression model, adjusting for these conditions, would be appropriate for comparing the intervention arms to the controls.

 This is the propensity score figure. 

>>>>>>>> 

• Provide also some descriptive statistics about the characteristics of the overall study participants in text form

>>>>>>>> 

Response: We added some descriptive statistics about the characteristics of the overall study participants in text form:

Of the total number of participants (65,425) in the intervention and control groups, 20,427 (30.9%) were White, 1,628 (2.4%) Asian, 25,615 (39.2%) Black or African American, 11,114 (17%) Hispanic, Latino or Spanish, and 6,641 (10.1%) did not provide racial and ethnic information. There were 10,056 (15.4%) participants had less than high school education, 16,947 (25.9%) with high school education, 16,307 (24.9%) had some college education, 19,384 (29.6%) with a college degree, 2,731 (4.1%) did not provide education information. 23,139 (35.4%) participants were between age 18-45, 29,466 (45%) between age 45-65, 9272 (14.2%) between age 65-75 and 3,548 (5.4%) participants older than 75. 6,828 (10.4%) completed any of the follow-up surveys, and of these, 6,725 (98.5%) completed all three follow-up surveys.

>>>>>>>> 

• Mention the completion rates in absolute terms in the results for all conditions. The figures are of low resolution and difficult to read. You could easily translate them into tables to make it easier to understand.

>>>>>>>> 

Response: We added the completion rates to a part of the fi

---

## [Decision Letter · Decision Letter 1]

5 Aug 2024

Improving Follow-Up Survey Completion Rates through Pilot Interventions in the All of Us Research Program: Results from a non-randomized intervention study

PONE-D-23-28732R1

Dear Dr. Cronin,

We’re pleased to inform you that your manuscript has been judged scientifically suitable for publication and will be formally accepted for publication once it meets all outstanding technical requirements.

Kind regards,

Christian von Wagner

Academic Editor

PLOS ONE

Additional Editor Comments (optional):

Thank you for your responses and I am pleased to accept the paper.

Reviewers' comments:

Reviewer's Responses to Questions

**Comments to the Author**

1. If the authors have adequately addressed your comments raised in a previous round of review and you feel that this manuscript is now acceptable for publication, you may indicate that here to bypass the “Comments to the Author” section, enter your conflict of interest statement in the “Confidential to Editor” section, and submit your "Accept" recommendation.

Reviewer #1: All comments have been addressed

2. Is the manuscript technically sound, and do the data support the conclusions?

Reviewer #1: Yes

3. Has the statistical analysis been performed appropriately and rigorously? 

Reviewer #1: Yes

4. Have the authors made all data underlying the findings in their manuscript fully available?

Reviewer #1: Yes

5. Is the manuscript presented in an intelligible fashion and written in standard English?

Reviewer #1: Yes

6. Review Comments to the Author

Reviewer #1: (No Response)

7. PLOS authors have the option to publish the peer review history of their article (what does this mean?). If published, this will include your full peer review and any attached files.

Reviewer #1: **Yes: **Nima Ghahari

---

## [Editor Report · Acceptance letter]

8 Aug 2024

PONE-D-23-28732R1 

PLOS ONE

Dear Dr. Cronin, 

I'm pleased to inform you that your manuscript has been deemed suitable for publication in PLOS ONE. Congratulations! Your manuscript is now being handed over to our production team.

Kind regards, 

on behalf of

Dr. Christian von Wagner 

Academic Editor

PLOS ONE